# How to Construct Polar Codes for Ring-LWE-Based Public Key Encryption

**DOI:** 10.3390/e23080938

**Published:** 2021-07-23

**Authors:** Jiabo Wang, Cong Ling

**Affiliations:** 1Beijing National Research Center for Information Science and Technology, Tsinghua University, Beijing 100084, China; 2Department of Electrical and Electronic Engineering, Imperial College London, London SW7 2AZ, UK; c.ling@imperial.ac.uk

**Keywords:** ring LWE, polar codes, public key encryption, error dependency, decryption failure rate

## Abstract

There exists a natural trade-off in public key encryption (PKE) schemes based on ring learning with errors (RLWE), namely: we would like a wider error distribution to increase the security, but it comes at the cost of an increased decryption failure rate (DFR). A straightforward solution to this problem is the error-correcting code, which is commonly used in communication systems and already appears in some RLWE-based proposals. However, applying error-correcting codes to those cryptographic schemes is far from simply installing an add-on. Firstly, the residue error term derived by decryption has correlated coefficients, whereas most prevalent error-correcting codes with remarkable error tolerance assume the channel noise to be independent and memoryless. This explains why only simple error-correcting methods are used in existing RLWE-based PKE schemes. Secondly, the residue error term has correlated coefficients leaving accurate DFR estimation challenging even for uncoded plaintext. It can be found in the literature that a tighter DFR estimation can effectively create a DFR margin. Thirdly, most error-correcting codes are not well designed for safety considerations, e.g., syndrome decoding has a nonconstant time nature. A code good at error correcting might be weak under a variety of attacks. In this work, we propose a polar coding scheme for RLWE-based PKE. A relaxed “independence” assumption is used to derive an uncorrelated residue noise term, and a wireless communication strategy, outage, is used to construct polar codes. Furthermore, some knowledge about the residue noise is exploited to improve the decoding performance. With the parameterization of NewHope Round 2, the proposed scheme creates a considerable DRF margin, which gives a competitive security improvement compared to state-of-the-art benchmarks. Specifically, the security is improved by 28.8%, while a DFR of $2^{-149}$ is achieved a for code rate pf 0.25, n=1024,q= 12,289, and binomial parameter k=55. Moreover, polar encoding and decoding have a quasilinear complexity $O(N\log_{2}N)$ and intrinsically support constant-time implementations.

## 1. Introduction

### 1.1. Error-Correcting for Ring-LWE-Based Public Key Encryption

The ring LWE (RLWE) problem was firstly introduced in 2010 [1], expanding on the classical version of the problem (i.e., LWE) introduced by Regev in [2]. Key establishment mechanisms based on RLWE, for example NewHope [3], are among the most attractive postquantum proposals. Their quantum security relies on the worst-case approximate shortest independent vector problem (SIVP), and they give better efficiency compared to plain LWE because of the ring structure. One topic of pressing importance is to refine such schemes for better efficiency and security. In this work, we focus on the issue of error correcting for RLWE-based public key encryption.

The key establishment based on RLWE is differentiated into two approaches regarding how to share the secret information. One is the “reconciliation-based approach” proposed by Ding et al. in [4] where Alice and Bob extract common information from the noisy secret with a robust extractor. However, Ding et al.’s reconciliation approach was observed to produce a biased secret, which cannot be used as a secret key. Peikert proposed another reconciliation method in [5], which directly produces a uniform secret key. In the initial version of NewHope [3], four-dimensional lattice codes and lattice quantization are used to design the reconciliation mechanism. To ease the reconciliation-based NewHope, Alkim et al. replaced the reconciliation with trivial repetition codes, which encode one bit of message into four coefficients of a polynomial in $R_{q}$. This gives rise to the second approach, i.e., the “encryption-based” approach. The encryption-based NewHope enjoys the same security properties and almost the same bandwidth requirement as the reconciliation-based one.

For most of the RLWE-based PKE schemes (e.g., NewHope, LIMA), one can enable a conversion from an indistinguishability under chosen plaintext attack (IND-CPA)-secure PKE to an indistinguishability under chosen ciphertext attack (IND-CCA)-secure scheme by applying the Fujisaki–Okamoto (FO) transform or its variants in a postquantum setting [6,7,8]. An obvious drawback of the FO transform is that it relies on the perfect correctness of the CPA-PKE scheme, which is not true due to the small residue noise after decryption. Although robust transforms against correctness errors were designed by Hofheinz et al. in [9], the advances of CCA attacks based on the decryption failure put current PQC candidates under threat. As a high-level description, the decryption failures in an RLWE-based PKE are somehow related to the secret information. If an adversary manages to observe enough failures and identify a correlation between failures and the secret, the security will be compromised. Therefore, the decryption failure rate (DFR) is a significant factor affecting the security level, and it should be precisely calculated. Most NIST submissions choose $2^{-128}$ as a target DFR.

D’Anvers et al. designed attacks exploiting the decryption failures, namely the “failure boosting” and “directional failure boosting”, in [10,11]. Using these techniques, an adversary can deliberately find “weak” ciphertexts that are more likely to trigger decryption failures. These failures are used to analyze the secret statistically. These attacks are verified on some basic versions of ring-/module-LWE-/LWR-based KEMs with comparable parameterization to NIST candidates, e.g., NTRUEncrypt, KYBER, and SABER, showing that the security of these KEM schemes is impacted under the proposed attacks assuming an unlimited number of decryption queries is allowed. In addition, Guo et al. proposed a CCA attack [12] targeting another RLWE-based NIST proposal, LAC. Furthermore, this novel attack exploits the high decryption failure rate of some ciphertexts caused by a certain weight property of the secret key. Though LAC was modified to resist those attacks, it was not selected as NIST’s finalist due to a variety of investigated and “hidden” attacks.

Some error-correcting codes, i.e., BCH and XEf, are used to improve the DFR in LAC and Round5, respectively [13,14]. This “unusual design” distinguishes them from other lattice-based schemes, e.g., NewHope uses repetition codes, while CRYSTALS-KYBER leaves the message uncoded. On the one hand, error-correcting codes can considerably increase the noise tolerance, improve the security, and save bandwidth. We noticed that the BCH, XEf, and repetition codes in above schemes are decoded according to hard decision metrics, e.g., the Hamming distance. We expect to see an impressive improvement if more powerful error-correcting codes (e.g., polar codes, low-density parity-check (LDPC) codes) are adopted. On the other hand, what comes with the usage of error-correcting codes is the risk of side-channel attacks based on timing/caching. Again, we take LAC as an example. Given polynomial $v^{\prime }=v-u\cdot s$ after decryption, the decoding of BCH codes proceeds in three steps: (1) recovering the codewords according to a hard decision metric, (2) calculating the syndrome, and (3) locating the errors if detected and correcting them. Obviously, the if⋯else statement in the last step is not constant-time.

Attempts to adapt modern error-correcting codes (e.g., LDPC) can be found in [15]. Experimental results were given to show how much LDPC codes can improve the DFR of NewHope Simple for a reasonably large enough binomial parameter *k*. Furthermore, the theoretical estimation of the upper bound on DFR using error-correcting codes was given based on an “independence” assumption claiming that the correlation between the coefficients of the residue noise $e\cdot t-s\cdot e^{\prime }+e^{\prime \prime }$ is negligible. However, these were actually not, and the soft decision decoding of LDPC assumes i.i.d. channels. The dependency among the noise coefficients is obvious in the vector representation of $e\cdot t-s\cdot e^{\prime }+e^{\prime \prime }$, i.e.,
(1)$$\left(\begin{array}{cccc} e_{0} & -e_{n-1} & \cdots & -e_{1}\\ e_{1} & e_{0} & \cdots & -e_{2}\\ \vdots & \vdots & \ddots & \vdots \\ e_{n-1} & e_{n-2} & \cdots & e_{0} \end{array}\right)t-\left(\begin{array}{cccc} s_{0} & -s_{n-1} & \cdots & -s_{1}\\ s_{1} & s_{0} & \cdots & -s_{2}\\ \vdots & \vdots & \ddots & \vdots \\ s_{n-1} & s_{n-2} & \cdots & s_{0} \end{array}\right)e^{\prime }+e^{\prime \prime }.$$

We have to be careful about the “independence” assumption: the assumption will overestimate (underestimate) DFR for schemes without (with) error-correction, and therefore underestimate (overestimate) the security. This “independence” assumption was relaxed by D’Anvers et al. in [16]. Specifically, the *i*-th coefficient of the noise term $e\cdot t-s\cdot e^{\prime }+e^{\prime \prime }$ is refined in the form of $c^{T}s+g$ where vector c is essentially determined by polynomials $e,e^{\prime }$, vector s by s,t, and scalar *g* by the *i*th coefficient of $e^{\prime \prime }$. They assumed $c^{T}s+g$ to be i.i.d. conditional on the $l_{2}$-norm of c and s. The DFR of LAC is interpreted as a weighted DFR averaged over all possible values of $∥s∥$, $∥c∥$. The ternary error terms in LAC make the calculation tractable. However, for a more general ring- (module)-LWE-based encryption with error terms drawn over Z, calculating the marginal distribution Pr$\left\{∥s∥\right\}$ and Pr$\left\{∥c∥\right\}$ is no longer trivial. In their prior work [17], they gave another assumption, namely the “Gaussian” assumption, to ease the calculation.

Song et al. interpreted NewHope as a digital communication system in [18]. At the transmitter’s end, binary message $m\in {\left\{0,1\right\}}^{256}$ is encoded as a codeword enc(m) by repeating *m*n/256 times. Then, enc(m) is modulated as a vector in ${\left\{0,\left\lfloor q/2\right\rfloor \right\}}^{n}$. At the receiver’s end, upon receiving $v^{\prime }=e\cdot t-s\cdot e^{\prime }+e^{\prime \prime }+\left\lfloor q/2\right\rfloor \cdot enc\left(m\right)$, the additive threshold decoder calculates $v_{i}^{\prime \prime }=\sum_{l=0}^{n/256-1}v_{i+256l}^{\prime }$ for i=0,1,⋯255 and recovers *m* by hard decision decoding. To analyze the DFR, one needs to take into account two types of dependencies in the noise term: (a) the dependency between the coefficients of $v^{\prime }$ conveying the same message bit of *m*, i.e., $v_{i+256l}^{\prime }$ for l=0,1,⋯,n/256−1; (b) the dependency between the n/4 coefficients of $v^{\prime \prime }$. In [18], $v_{i}^{\prime \prime }$ was elegantly written in the form of $v_{i}^{\prime \prime }=\sum_{j=0}^{511}W_{i,j}+\sum_{l=0}^{n/256-1}n_{i+256l}$ as was the sum of 512 i.i.d. random variables $W_{i,j}$ and n/256 i.i.d. random variables $n_{i+256l}$ for any fixed *i*. Therefore, the first-type dependency was addressed. As for the second type, Song et al. proved the error term $v_{i}^{\prime \prime }$ to be identically distributed for any i=0,1,⋯,n/4, and therefore gave a union bound on the DFR. Consequently, a tighter upper bound on the DFR is derived, which is less than $2^{-418}$ for n=1024 and $2^{-399}$ for n=512 (The NewHope submission claims to have an upper bound on DFR to be $2^{-216}$ for n=1024 and $2^{-213}$ for n=512). The improved DFR margin enhances the security level without any changes to the original protocol.

The motivation of this work was to investigate how to handle the dependency of RLWE-based PKE and how to adapt modern error-correcting codes to it. We sought a security improvement using the derived DFR margin. A concurrent work can be found in [19], where canonical embedding was employed to derive i.i.d. fading channels with channel state information (CSI) available to the recipient and polar codes were constructed. However, in reality, we do not expect to engagein canonical embedding because we can: (a) spare ourselves the trouble of switching between the canonical and polynomial representation; (b) avoid the error tolerance loss due to the tailored constellation diagram as [19] illustrated; (c) make the overall scheme comply with the most popular and practical RLWE-based PKE framework where we only deal with integers on the interval [0,q).

### 1.2. Contribution

The contribution of this paper is as follows.

We formulated the RLWE-based PKE as an i.i.d. mod 2Z additive Gaussian noise channel with channel state information (CSI) available to the receiver under a relaxed “independence” assumption;(a)Given the residue noise term $e\cdot t-s\cdot e^{\prime }+e^{\prime \prime }$, we formulated the RLWE-based PKE as a mod 2Z additive Gaussian noise channel within exactly one code block. We assumed the mod 2Z additive Gaussian channel to be independent under a relaxed assumption compared to the one in [15];(b)Alice, the decoder, can considerably improve the DFR by exploiting the advantage that the polynomials *e* and *s* are generated on her side and she can figure out the precise distribution of the Gaussian noise;We employed a telecommunication-engineering strategy, namely outage, to construct polar codes for RLWE-based PKE. The encoding and decoding routines allow quasilinear (i.e., $\left(N\log_{2}N\right)$) and constant-time implementations. Experimental results and theoretical estimation of DFR are also given. Specifically, we derived a new DFR of $2^{-149}$ by SC decoding for NewHope parameters q= 12,289, n=1024 and code rate = 0.25 and a larger central binomial parameter k=55. The DFR margin enabled us to improve the security by 28.8% while keeping the target DFR of $2^{-140}$ (as is the benchmark in the work of [15,18]) achievable.

### 1.3. Roadmap

This paper is organized as follows. A review of the ring-LWE-based public key encryption and some basics of channel models and polar codes can be found in Section 2. The problem formulation and methodology are introduced in Section 3. In Section 3.1, we explain how to formulate a typical RLWE-based PKE scheme as a mod 2Z channel with additive Gaussian noise. A relaxed “independence” assumption is used to derive i.i.d. channels. We explain the soundness of the proposed scheme in Section 3.2 and demonstrate how to construct and decode polar codes explicitly in Section 3.3. In Section 4, we analyze the DFR theoretically and experimentally when polar decoding (SC decoding) is applied. We, in Section 5, discuss the security improvement, the constant-time implementation, and communication overhead increase by polar encoding and decoding. We conclude this paper in Section 6.

## 2. Preliminaries

### 2.1. Ring-LWE Public Key Encryption Scheme

The public key encryption scheme based on ring-LWE was first described in [20] and formally defined in a subsequent work [21]. We use the “informal” definition of ring-LWE given in [20], as it then became the most prevalent version in implementations, e.g., NewHope [22] and Peikert’s KEM [5]. The scheme is parameterized by an integer modulus *q*, dimension *n*, a power of two, and a ring of integers $R:=\frac{Z[X]}{x^{n}+1}$ and its quotient ring $R_{q}:=R/qR$. We define an error distribution χ over *R*. We take the example of NewHope and define sampling from χ to be sampling each coefficient of a polynomial in *R* from a discrete Gaussian over Z. The scheme proceeds as follows:Alice firstly samples $a\in R_{q}$ uniformly at random, then she samples a secret key *s* together with an error *e* according to χ. She publishes as the public key a ring-LWE sample $\left(a,b\right)=\left(a,a\;\cdot s+e\;\mathrm{mod}\;q\right)\in R_{q}\times R_{q}$;Bob encrypts a message $m\in {\left\{0,1\right\}}^{n}$ as $\left(c_{1},c_{2}\right)=\left(a\cdot t+e^{\prime }\;\mathrm{mod}\;q,b\cdot t+e^{\prime \prime }+\left\lfloor \frac{q}{2}\right\rfloor \cdot m\;\mathrm{mod}\;q\right)$, where $e^{\prime },e^{\prime \prime },t$ are sampled independently from χ;Alice decrypts using *s* by computing $d:=c_{2}-c_{1}\cdot s=\left\lfloor \frac{q}{2}\right\rfloor \cdot m+e\cdot t-s\cdot e^{\prime }+e^{\prime \prime }$.

Alice then recovers the message *m* by decoding: if the *i*th coordinate of *d* is closer to zero than ⌊q/2⌋, Alice assumes the *i*th coordinate of *m* was zero, otherwise she assumes it was one. We observe a few key facts about this scheme that we need for our work. Firstly, although its formal security proof may be found in [21], the main idea is that $b,c_{1}$, and $c_{2}$ leak no information about the secret *s* and the plaintext *m* because they are ring-LWE samples, which are assumed to be pseudorandom by the hardness of the ring-LWE decision problem. Therefore, one could alternate the encoding term $\lfloor \frac{q}{2}\rfloor \cdot m$ without affecting security, as long as the encoding is independent of the actualization of the variables $s,e,e^{\prime },e^{\prime \prime },t$. We use this fact implicitly while constructing polar codes in the sequel. Secondly, we observe that Alice knows the actualization of *s* and *e*, and so may use these for decoding.

### 2.2. Channel Models

In wireless communications, the additive white Gaussian noise (AWGN) channel is the most primary and frequently used model to characterize how noises interfere with the channel input. A typical discrete-time AWGN channel is defined as:$$y_{i}=x_{i}+z_{i},\;\;i=1,\cdots ,N,$$
where $x_{i}\in R$ is the channel input, $y_{i}\in R$ is the channel output, and $z_{i}$ is an additive white Gaussian noise, and there are *N* time slots in total. Ideally, these variables are independent in different time slots indicated by subscript *i*. A fading channel arises due to a time-varying attenuation of signal quality caused by either the propagation environment or by the movement of the transmitter/receiver. We consider a fading channel model *W* as:$$y_{i}=h_{i}x_{i}+z_{i},\;\;\;i=1,\cdots ,N,$$
where $h_{i}$ is the channel gain and $z_{i}$ is additive white Gaussian noise. Denote by $T_{c}$ the coherence interval of a fading channel *W*. In the context of a fading channel with memory, the channel gain $h_{i}$ is believed to be a constant within one coherence interval and varies independently as the next coherence interval approaches. The realization of $h_{i}$ is called channel state information (CSI), and the distribution of $h_{i}$ is called channel distribution information (CDI). In the special case of $T_{c}=1$, channel *W* is referred to as an identically independently distributed (i.i.d.) fading channel. The design and performance of error-correcting codes for i.i.d. fading channels with/without CSI is well studied [23,24,25,26,27].

How to design $x_{i}$ to reliably transmit information at the highest rate via a specific channel has been widely and comprehensively studied over the past decades. A branch of this study is to construct capacity-achieving lattice codes for an AWGN channel and its fading variants [28,29,30,31]. At the transmitter’s end, lattice coding maps binary codes to a constellation diagram in Euclidean space, called lattice modulation. At the recipient’s end, the decoder recovers the binary codes by the bounded distance decoding or preferably maximum likelihood decoding for better performance. This leads to the definition of mod Λ channel and $\Lambda /\Lambda^{\prime }$ channel where Λ is a lattice and $\Lambda^{\prime }$ is a sublattice of Λ. We omit the formal definition here, but give an example of a mod Z channel and a Z/2Z channel, which will be used in Section 3.1.

**Example** **1.**
*A mod Z channel is an additive white Gaussian noise (AWGN) channel with input restricted to a∈V(Z) where V(Z) is the fundamental region (A fundamental region of a lattice Λ is a region that includes one and only one point from each coset of Λ in $R^{n}$. Algebraically, V(Λ) is a set of coset representatives for all the cosets of Λ in $R^{n}$, e.g., we can define V(Z) to be [0,1), but not necessarily to be the fundamental Voronoi cell [−0.5,0.5).) of Z. At the receiver’s end, there is a mod V(Z) operation giving the equivalent channel output as:*
$$y=a+n\;mod\;Z=(a+n^{\prime })\;mod\;Z,$$
*where n is the AWGN noise and $n^{\prime }=n\;mod\;Z$.*


**Example** **2.**
*A Z/2Z channel is an AWGN channel with input restricted to r∈(Z+a)∩V(2Z) for some offset a∈R. At the receiver’s end, the equivalent channel output is:*
$$y=r+n\;mod\;2Z=r+n^{\prime }\;mod\;2Z.$$

*It can be viewed as a mod 2Z channel with input restricted to a set of elements of Z+a that fall in V(2Z).*


### 2.3. Polar Codes for BDMS Channels

Polar codes, introduced by Arıkan in [32], are linear block codes of length $N=2^{n}$ for a positive integer *n* that achieves the capacity of any binary input discrete memoryless symmetric (BDMS) channels asymptotically (In fact, the generalizations of polar codes are extended to arbitrary code length and a large class of channels.). We firstly recall some basics of polar coding for a BDMS channel. Given a BDMS channel *W*, there are two commonly used metrics in information theory to measure the quality of *W*: the mutual information (The maximum mutual information over all possible channel input distributions is the channel capacity.) and the reliability.

**Definition** **1** (Mutual information of BDMS channels)**.** 
*The mutual information I(W) of a channel W is the maximum rate at which information can be successfully transmitted from the transmitter to the receiver. For a BDMS channel W:X→Y, I(W)∈[0,1] is defined as:*
$$\begin{array}{c} I\left(W\right)≜\sum_{y\in Y}\sum_{x\in X}\frac{1}{2}W\left(y|x\right)\log_{2}\frac{W(y|x)}{\frac{1}{2}W\left(y|0\right)+\frac{1}{2}W\left(y|1\right)}. \end{array}$$


Here, we use the definition of symmetric mutual information assuming a uniform channel input, which is also the capacity of the BDMS channel. We use the notations I(W) and I(Y;X) interchangeably to denote the mutual information of *W*.

**Definition** **2**(Bhattacharyya parameter of BDMS channels). *The Bhattacharyya parameter Z(W) is a measure of channel reliability. For a BDMS channel W, Z(W)∈[0,1] is defined as:*
$$\begin{array}{c} Z\left(W\right)≜\sum_{y\in Y}\sqrt{W(y|0)W(y|1)}. \end{array}$$
*A small Z(W) indicates a more reliable channel, while a large Z(W) implies a channel with more inferences.*


The capacity-achieving nature of polar codes arises from the so-called channel polarization phenomenon as a result of recursive applications of Arıkan’s transform to identical *W*s and their synthesized derivatives. The overall recursive transform can be performed in a channel-combining phase and a channel-splitting phase. In the channel-combining phase, a linear transformation defined as $X^{1:N}=U^{1:N}G_{N}$ is performed on a vector $U^{1:N}\in X^{1:N}$ over GF(2), where $G_{N}=B_{N}{\left[\begin{array}{cc} 1 & 0\\ 1 & 1 \end{array}\right]}^{⊗n}$. $B_{N}$ is a permutation matrix: if $U^{\prime 1:N}=U^{1:N}B_{N}$ and $n=\log_{2}N$, the $i^{\prime }=\left({\left(b_{n},\cdots ,b_{2},b_{1}\right)}_{2}+1\right)$-th coordinate of $U^{\prime 1:N}$ is the $i=({\left(b_{1},b_{2},\cdots ,b_{n}\right)}_{2}+1)$-th coordinate of $U^{1:N}$. By taking $X^{1:N}$ as the raw input of *W*, one derives a combined channel $W_{N}:X^{1:N}\to Y^{1:N}$ with a transition probability of:(2)$$\begin{array}{c} W_{N}\left(y^{1:N}|u^{1:N}\right)=\prod_{i\in \{1,\cdots ,N\}}W\left(y^{\left(i\right)}|x^{\left(i\right)}={\left(u^{1:N}G_{N}\right)}_{i}\right), \end{array}$$
where ${\left(\cdot \right)}_{i}$ denotes *i*-th coordinate. Since $G_{N}$ induces a one-to-one mapping between $U^{1:N}$ and $X^{1:N}$, the mutual information of $W_{N}$ is:(3)$$\begin{array}{c} I\left(W_{N}\right)=I\left(Y^{1:N};U^{1:N}\right)=NI\left(W\right). \end{array}$$
In the channel-splitting phase, $W_{N}$ is further split back into *N* synthesized channels $W_{N}^{\left(i\right)}:X\to Y^{N}\times X^{i-1}$ whose transition probability is defined by:(4)$$\begin{array}{cc} W_{N}^{\left(i\right)}\left(y^{1:N},u^{1:i-1}|u^{\left(i\right)}\right) & =\sum_{U^{i+1:N}\in X^{N-i}}\frac{1}{2^{N-1}}W_{N}\left(Y^{1:N}|U^{1:N}\right). \end{array}$$

We now demonstrate how to perform Arıkan’s transform. We begin with the transform on two i.i.d. BDMS channels W:{0,1}→Y as shown in Figure 1. Let $X^{1:2}=\left(X^{\left(1\right)},X^{\left(2\right)}\right)\in {\left\{0,1\right\}}^{2}$ be the raw input vector of two *W* and $X^{1:2}=\left(Y^{\left(1\right)},Y^{\left(2\right)}\right)\in Y^{2}$ be the raw channel output vector. Denote by $U^{1:2}=\left(X^{\left(1\right)},X^{\left(2\right)}\right)\in {\left\{0,1\right\}}^{2}$ the message vector. The symbol ⊕ indicates a mod-2 operation.

At the channel-combining stage, the message vector $U^{1:2}$ is transformed into $X^{1:2}=U^{1:2}G_{2}\;\mathrm{mod}\;2$ where $G_{2}=\left[\begin{array}{cc} 1 & 0\\ 1 & 1 \end{array}\right]$. The two parallel *W*s are seen as a combination channel $W_{2}:{\left\{0,1\right\}}^{2}\to Y^{2}$. Since there exists a bijection between $U^{1:2}$ and $X^{1:2}$, the transition probability of $W_{2}$ is:$$W_{2}\left(y^{1:2}|u^{1:2}\right)=W\left(y^{\left(1\right)}|u^{\left(1\right)}⊕u^{\left(2\right)}\right)W\left(y^{\left(2\right)}|u^{\left(2\right)}\right).$$
The channel capacity of $W_{2}$ and *W* satisfies:(5)$$\begin{array}{c} I\left(W_{2}\right)=2I\left(W\right). \end{array}$$

At the channel-splitting stage, the combination channel $W_{2}$ is split into two synthesized channels $W_{2}^{\left(1\right)}:\left\{0,1\right\}\to Y^{2}$ and $W_{2}^{\left(2\right)}:\left\{0,1\right\}\to Y^{2}\times \left\{0,1\right\}$. To be specific, channel $W_{2}^{\left(1\right)}$ takes $U^{\left(1\right)}$ as the only input and gives $Y^{\left(1\right)},Y^{\left(2\right)}$ as the output. As for channel $W_{2}^{\left(2\right)}$, it takes $U^{\left(2\right)}$ as the only channel input and gives $Y^{\left(1\right)},Y^{\left(2\right)}$ and $U^{\left(1\right)}$ as the channel output. The channel transition probabilities of $W_{2}^{\left(1\right)}$ and $W_{2}^{\left(2\right)}$ are:(6)$$\begin{array}{cc} W_{2}^{\left(1\right)}\left(y^{1:2}|u^{\left(1\right)}\right) & =\sum_{u^{\left(2\right)}\in \left\{0,1\right\}}\frac{W_{2}\left(y^{1:2}|u^{1:2}\right)\cdot P\left(u^{\left(1:2\right)}\right)}{P(u^{\left(1\right)})}\\ \; & \overset{\left(a\right)}{=}\frac{1}{2}\sum_{u^{\left(2\right)}\in \left\{0,1\right\}}W\left(y^{\left(1\right)}|u^{\left(1\right)}⊕u^{\left(2\right)}\right)W\left(y^{\left(2\right)}|u^{\left(2\right)}\right) \end{array}$$
and:(7)$$\begin{array}{cc} W_{2}^{\left(2\right)}\left(y^{1:2},u^{\left(1\right)}|u^{\left(2\right)}\right) & =\frac{W_{2}\left(y^{1:2}|u^{1:2}\right)\cdot P\left(u^{\left(1:2\right)}\right)}{P(u^{\left(2\right)})}\\ \; & \overset{\left(b\right)}{=}\frac{1}{2}W\left(y^{\left(1\right)}|u^{\left(1\right)}⊕u^{\left(2\right)}\right)W\left(y^{\left(2\right)}|u^{\left(2\right)}\right). \end{array}$$

Note that the equalities (a)(b) are derived because $U^{\left(1\right)}$, $U^{\left(2\right)}$ are i.i.d. and they are uniformly distributed over {0,1}. More generally, a proposition follows to show the relation between $(W_{N}^{\left(i\right)}$, $W_{N}^{\left(i\right)})$ and $(W_{2N}^{\left(2i-1\right)}$, $W_{2N}^{\left(2i\right)})$.

**Proposition** **1**([32])**.** *For i=1,⋯,N,*
(8)$$\begin{array}{c} W_{2N}^{\left(2i-1\right)}\left(y^{1:2N},u^{1:2i-2}|u^{\left(2i-1\right)}\right)\\ =\frac{1}{2}\sum_{u^{\left(2i\right)}}W_{N}^{\left(i\right)}\left(y^{1:N},u_{o}^{1:2i-2}⊕u_{e}^{1:2i-2}|u^{\left(2i-1\right)}⊕u^{\left(2i\right)}\right)\cdot W_{N}^{\left(i\right)}\left(y^{N+1:2N},u_{e}^{1:2i-2}|u^{\left(2i\right)}\right) \end{array}$$
(9)$$\begin{array}{c} W_{2N}^{\left(2i\right)}\left(y^{1:2N},u^{1:2i-1}|u^{\left(2i\right)}\right)\\ =\frac{1}{2}W_{N}^{\left(i\right)}\left(y^{1:N},u_{o}^{1:2i-2}⊕u_{e}^{1:2i-2}|u^{\left(2i-1\right)}⊕u^{\left(2i\right)}\right)\cdot W_{N}^{\left(i\right)}\left(y^{N+1:2N},u_{e}^{1:2i-2}|u^{\left(2i\right)}\right), \end{array}$$
*where $u_{o}^{1:2i-2}$ and $u_{e}^{1:2i-2}$ indicate a subvector of $u^{1:2i-2}$ of odd and even indices, respectively.*

It was proven in [32] that Arıkan’s transform preserves the mutual information in the sense that:$$I\left(W_{N}\right)=NI\left(W\right)=\sum_{i\in \{1,\cdots ,N\}}I\left(W_{N}^{\left(i\right)}\right).$$

More importantly, the quality of the synthesized channels polarizes asymptotically as the recursion proceeds.

**Theorem** **1**(Channel polarization of mutual information [32]). *For any BDMS channel W, the synthesized channels $W_{N}^{\left(i\right)}$ polarize in the sense that, for any fixed δ∈(0,1), as N goes to infinity through powers of two, the fraction of indices i∈{1,⋯,N} for which $I\left(W_{N}^{\left(i\right)}\right)\in \left(1-\delta ,1\right]$ goes to I(W) and the fraction for which $I\left(W_{N}^{\left(i\right)}\right)\in \left[0,\delta \right)$ goes to 1−I(W).*

The channel polarization theorem from above can also be stated in the metric of the Bhattacharyya parameter by replacing $I(W_{N}^{\left(i\right)})$ by $Z(W_{N}^{\left(i\right)})$.

For any desired transmission rate R<I(W), we can partition {1,⋯,N} into a subset A and its complement $A^{C}$ such that (i) |A|=⌊NR⌋ and (ii) for any i∈A and $j\in A^{C}$, $Z\left(W_{N}^{\left(i\right)}\right)\leq Z\left(W_{N}^{\left(j\right)}\right)$. Denote by $G_{N}\left(A\right)$ (resp. $G_{N}\left(A^{C}\right)$) the rows of $G_{N}$ indexed by A (resp. $A^{C}$). Given the most reliable ⌊NR⌋ channels indexed by A, one can construct polar codes following the encoding rule:(10)$$\begin{array}{c} X^{1:N}=U_{A}G_{N}\left(A\right)⊕U_{A^{C}}G_{N}\left(A^{C}\right), \end{array}$$
where $U_{A}$ is the useful information vector of length ⌊NR⌋ and $U_{A^{C}}$ is a predetermined vector, named frozen bits, known to both the encoder and decoder, e.g., $U_{A^{C}}=0$. In this manner, the useful information is transmitted via the most reliable synthesized channels. A question may arise about how to efficiently calculate $Z(W_{N}^{\left(i\right)})$. A brief review can be found in Section 2.4 and Section 3.3. As a high-level description, calculating $Z(W_{N}^{\left(i\right)})$ according to Definition 2 for a BDMS channel with a large or even continuous output alphabet is not easy because the output alphabet of the synthesized channel $W_{N}^{\left(i\right)}$ increases exponentially with a factor of $\log_{2}N$. One solution to handle this problem is to firstly construct an approximate channel $W^{\prime }$ of *W* using a degrading/upgrading technique such that $W^{\prime }$ has a countable output alphabet of a size no greater than μ and only minor and traceable capacity loss [33]. Then, one applies Arıkan’s transform recursively to $W^{\prime }$, deriving synthesized channels as Proposition 1 indicates. At each recursion, one applies a merging technique to approximate the synthesized channels such that the approximation is stochastically degraded with the original one and has an output alphabet no greater than a predetermined value (e.g., ν) [34]. In this way, one can finally derive an approximation of $W_{N}^{\left(i\right)}$ with an output alphabet no larger than ν and negligible capacity difference. Now, one is able carry out the encoding as in Formula (Equation 10).

The successive cancellation (SC) decoder is the initial decoding algorithm for polar codes. It gives an estimation of $u^{\left(i\right)}$, the *i*-th coordinate of $U^{1:N}$, in the natural order of *i*. Given a polar code parameterized by code length *N*, information set A, and frozen bits $U_{A^{C}}$, one can derive the recovered message ${\bar{u}}^{\left(i\right)}$ of $u^{\left(i\right)}$ in sequential order of index *i* according to the decoding rule specified as:(11)$$\begin{array}{c} {\bar{u}}^{\left(i\right)}=\left\{\begin{array}{cc} u^{\left(i\right)} & i\in A^{C},\\ 0 & L_{N}^{\left(i\right)}\left(y^{1:N},{\bar{u}}^{1:i-1}\right)\geq 1\;\mathrm{and}\;i\in A,\\ 1 & \mathrm{otherwise}, \end{array}\right. \end{array}$$
where ${\bar{u}}^{1:i-1}$ is the estimation of $u^{1:i-1}$ recovered before ${\bar{u}}^{\left(i\right)}$ and $L_{N}^{\left(i\right)}\left(y^{1:N},{\bar{u}}^{1:i-1}\right)$ is the likelihood ratio function defined as:$$\begin{array}{c} L_{N}^{\left(i\right)}\left(y^{1:N},{\bar{u}}^{1:i-1}\right)=\frac{W_{N}^{\left(i\right)}\left(y^{1:N},{\bar{u}}^{1:i-1}|u^{\left(i\right)}=0\right)}{W_{N}^{\left(i\right)}\left(y^{1:N},{\bar{u}}^{1:i-1}|u^{\left(i\right)}=1\right)}. \end{array}$$

The computational complexity of SC decoding, as is dominated by the recursive calculation of $L_{N}^{\left(i\right)}$ (see Appendix A), is $O(N\log_{2}N)$.

Denote by $P_{e}$ the average probability of block decoding error. As a result of polar encoding and SC decoding, it was proven in [32] that $P_{e}$ is upper bounded as follows.

**Theorem** **2**(Decoding performance [32]). *For any BDMS channel W and any choices of parameter (N,R,A),*
$$\begin{array}{c} P_{e}\leq \sum_{i\in A}Z\left(W_{N}^{\left(i\right)}\right). \end{array}$$

### 2.4. Channel Degradation and Upgradation

The construction of polar codes can be addressed if all the Bhattacharyya parameters of synthesized channels can be efficiently calculated. In [32], an efficient solution to compute $Z(W_{N}^{\left(i\right)})$ for binary erasure channels (BEC) was given, while it was suggested to use the Monte Carlo method to deal with more general BDMS channels. R. Mori and T. Tanaka made an attempt to solve this problem for arbitrary binary input memoryless symmetric (BMS) channels using the density evolution [35,36,37] of belief propagation (BP) decoding. However, they also mentioned that it was unclear how to handle the computational efficiency when the code length *N* was large and the requirement for precision was high. In [33], a quantization method was proposed to construct a degraded and upgraded approximation of a general BMS channel. If the degraded or upgraded relation exists, one can approximate $Z(W_{N}^{\left(i\right)})$ efficiently.

**Definition** **3**(Degraded and upgraded channel [33]). *A channel Q:X→Z is (stochastically) degraded with respect to a channel W:X→Y if there exists a channel P:Y→Z such that:*
$$Q\left(z|x\right)=\sum_{y\in Y}W\left(y|x\right)P\left(z|y\right)$$
*for all z∈Z and x∈X. We denote by Q⪯W the relation that Q is degraded with respect to W. Conversely, we denote by $Q^{\prime }⪰W$ the relation that $Q^{\prime }$ is upgraded with respect to W if there exists a channel $Q^{\prime }:X\to Z^{\prime }$ and a channel $P:Z^{\prime }\to Y$ such that:*
$$W\left(y|x\right)=\sum_{z^{\prime }\in Z^{\prime }}Q^{\prime }\left(z^{\prime }|x\right)P\left(y|z^{\prime }\right)$$
*for y∈Y and x∈X.*

Moreover, the synthesized channels of $Q,W,Q^{\prime }$ under Arıkan’s transform also fulfill the channel degradation and upgradation relation.

**Lemma** **1**(Restatement of Lemma 4.7 in [38]). *Given BMS channels W,Q, and $Q^{\prime }$, we denote by $W_{N}^{\left(i\right)}$, $Q_{N}^{\left(i\right)}$, and $Q_{N}^{\prime }\left(i\right)$ for i∈[1,N] the synthesized channels obtained by Arıkan’s transformation. If $Q^{\prime }⪰W⪰Q$ for all i, then $Q_{N}^{\prime }\left(i\right)⪰W_{N}^{\left(i\right)}⪰Q_{N}^{\left(i\right)}$.*

If the channel degradation or upgradation relation is set up, their channel capacity, reliability, and error probability will be related as follows.

**Lemma** **2**([33]). *Let W be a BMS channel, and suppose there exists another channel Q such that Q⪯W. Then:*
$$\begin{array}{cc} C(Q) & \leq C(W),\\ Z(Q) & \geq Z(W),\\ P_{e}\left(Q\right) & \geq P_{e}\left(W\right). \end{array}$$
*The inequality will reverse if we replace “degraded” by “upgraded”.*


## 3. Materials and Methods

### 3.1. RLWE-Based PKE Channel Model with Outage

In the field of telecommunication, a signal outage occurs if the signal power at the receiver’s end falls below a threshold, which is related to the minimum signal-to-noise ratio (SNR) acceptable to the communication performance. The outage probability is defined as the probability with which signal outage occurs. The analysis of outage probability is of great importance to estimate fading capacities in a fading environment. A typical example is the outage estimation for fading multiple-input and multiple-output (MIMO) channels [39,40].

We already gave an RLWE-based PKE instance in Section 2. We now consider the problem of decoding the message *m* given the polynomial:(12)$$y=\left\lfloor \frac{q}{2}\right\rfloor \cdot m+e\cdot t-s\cdot e^{\prime }+e^{\prime \prime }\;\mathrm{mod}\;R_{q},$$
where e·t and $s\cdot e^{\prime }$ are polynomial multiplications in $Z\left[x\right]/\left(1+x^{n}\right)$. It can be written in vector form as:(13)$$\left\lfloor \frac{q}{2}\right\rfloor m+\left(\begin{array}{cccc} e_{0} & -e_{n-1} & \cdots & -e_{1}\\ e_{1} & e_{0} & \cdots & -e_{2}\\ \vdots & \vdots & \ddots & \vdots \\ e_{n-1} & e_{n-2} & \cdots & e_{0} \end{array}\right)t-\left(\begin{array}{cccc} s_{0} & -s_{n-1} & \cdots & -s_{1}\\ s_{1} & s_{0} & \cdots & -s_{2}\\ \vdots & \vdots & \ddots & \vdots \\ s_{n-1} & s_{n-2} & \cdots & s_{0} \end{array}\right)e^{\prime }+e^{\prime \prime }\mathrm{mod}\;q.$$

Since the receiver knows matrices E,S and we observe that the norm of each row of E,S stays the same within one code block, the channel model of RLWE-based PKE can be described in a fading channel form as:(14)$$Y_{i}=\left\lfloor \frac{q}{2}\right\rfloor m_{i}+H*Z_{i},\;\mathrm{mod}\;q,\;i=1,\cdots n,$$
where $m_{i}\in \left\{0,1\right\}$, $Z_{i}\leftarrow N\left(0,r^{2}\right)$ and the channel gain *H* is $H=\sqrt{1+\sum_{1}^{n}e_{i}^{2}+\sum_{1}^{n}s_{i}^{2}}$ where $e_{i}$ and $s_{i}$ are coefficients of polynomials *e* and *s* for i∈[n], respectively. Note that we assume the error distribution χ to be a normal distribution $N(0,r^{2})$ for the convenience of analysis. A similar setting can be found in [41] where χ is defined on R/[0,q).

*Independence assumption:* Taking a close look at the channel model in Formula (Equation 14), we derive a group of *n* identically distributed channels rather than i.i.d. channels because every $Z_{i}$ is related to every coordinate of t and $e^{\prime }$. To apply polar codes to the encoding and decoding step, we assume that the correlation between the $Z_{i}$s are negligible and will not affect the decoding performance, as is a common assumption when applying modern error-correcting codes to RLWE-based PKE [15].

Now, we denote by ϵ∈(0,1) the outage probability and denote by $H_{\epsilon }$ the threshold such that $\mathrm{Pr}\{H>H_{\epsilon }\}=\epsilon$. Unlike in a telecommunication system where the uncertainty of channel gain would introduce difficulties in estimating the outage probability, in our RLWE channel, how the fading behaves is clearly known to the receiver. In the RLWE-based PKE instance in Section 2, both participants of the PKE process know the distribution of *H*. Moreover, Alice, who plays the role of the receiver in telecommunication, precisely knows the value of *H*, i.e., the channel state information. Examples of how $H_{\epsilon }$ is defined can be seen in Figure 2 where ϵ=0.01 and *r* is the parameter of normal distribution $N(0,r^{2})$.

The revised public key encryption proceeds as follows:The key generation step is the same as the RLWE-based PKE instance in Section 2;At the encryption step, Bob takes the RLWE channel as a mod2Z additive Gaussian channel (To be precise, it is a $\lfloor \frac{q}{2}\rfloor Z/qZ$ channel with additive Gaussian noise $N(0,r^{2}H^{2})$ or, equivalently, a Z/2Z channel. To ease the notation, we instead use the mod2Z channel with input restricted to {0,1}. The two channels are statistically equivalent.) with the Gaussian distribution to be $N(0,r^{2}H_{\epsilon }^{2})$. Then, he constructs polar codes of code length N=n for this channel as described in Section 2.3 and carries out encryption as normal;At the decryption step, Alice firstly calculates $H=\sqrt{1+\sum_{1}^{n}e_{i}^{2}+\sum_{1}^{n}s_{i}^{2}}$. If $H>H_{\epsilon }$, Alice goes back to the key generation step, and the whole process is restarted; otherwise, she decrypts and carries out SC decoding for the mod 2Z channel with additive Gaussian noise $N(0,r^{2}H^{2})$. (An explicit illustration of polar encoding and decoding is given in Section 3.3.)

### 3.2. The Soundness and Security of the Proposed Scheme

In the above revised RLWE-based PKE scheme, we construct polar codes for a mod 2Z channel with additive noise $N(0,r^{2}H_{\epsilon }^{2})$, then apply the codes to a mod 2Z channel with additive noise $N(0,r^{2}H^{2})$ where $H\leq H_{\epsilon }$. The soundness is guaranteed by the channel degradation relationship between the two channels.

**Lemma** **3.**
*If $\sigma_{1}<\sigma_{2}$, the $N(0,\sigma_{2}^{2})\;\mathrm{mod}\;2Z$ channel is degraded with respect to the $N(0,\sigma_{1}^{2})\;\mathrm{mod}\;2Z$ channel.*


**Proof.** Suppose the channel input is *X*, and let $N_{1}\leftarrow N\left(0,\sigma_{1}^{2}\right)$ and $N_{2}\leftarrow N\left(0,\sigma_{2}^{2}\right)$ be additive noises. We also define an auxiliary additive noise denoted by $N_{aux}$, which is drawn from $N(0,\sigma_{2}^{2}-\sigma_{1}^{2})$. At the recipient’s end, the channel output after the mod2Z operation is $Y=\left(X+N_{2}\right)\;\mathrm{mod}\;2Z=\left(\left(X+N_{1}\right)\;\mathrm{mod}\;2Z+N_{aux}\right)\;\mathrm{mod}\;2Z$. As a result, $N_{2}\;\mathrm{mod}\;2Z$ can be interpreted as a concatenation of $N_{1}\;\mathrm{mod}\;2Z$ and $N_{aux}\;\mathrm{mod}\;2Z$. The proof is complete according to the definition of channel degradation as in Section 2.4. □

We now have the degradation relation between the channel models Bob and Alice have access to, i.e., $N(0,H_{\epsilon }^{2}r^{2})$ mod $2Z⪯N(0,H^{2}r^{2})$ mod 2Z. Recall that Lemma 2 quantitatively shows from what aspect one channel is degraded to the other and Lemma 1 shows that Arıkan’s transform preserves the channel degradation relation. Meanwhile, constructing polar codes is performed by selecting the most reliable synthesized channels to convey the message. As a result, the polar code customized for $N(0,H_{\epsilon }^{2}r^{2})$ mod 2Z is a subcode of the polar codes customized for the channel $N(0,H^{2}r^{2})$ mod 2Z. A similar technique by which one can construct a polar code for a degraded channel and apply it to the channel in reality can be found in [30]. The explicit polar encoding and SC decoding processes are given in Section 3.3.

**Definition** **4**(CPA indistinguishability experiment [42]). *Consider a public key encryption scheme Π=(Gen,Enc,Dec) and an adversary A; the chosen plaintext attack (CPA) indistinguishability experiment ${PubK}_{A,\Pi }^{cpa}\left(n\right)$ is defined as follows:*
*1* *$Gen(1^{n})$ is run to obtain keys (pk,sk);**2* *Adversary A is given pk, as well as oracle access to $Enc_{pk}\left(\cdot \right)$. The adversary outputs a pair of messages $m_{0},m_{1}$ of the same length (these messages must be in the plaintext space associated with pk);**3* *A random bit b←{0,1} is chosen, and then, a ciphertext $c\leftarrow Enc_{pk}\left(m_{b}\right)$ is computed and given to A. We call c the challenge ciphertext;**4* *A continues to have access to ${Enc}_{pk}\left(\cdot \right)$ and outputs a bit $b^{\prime }$;**5* *The output of the experiment is defined to be 1 if $b^{\prime }=b$, and 0 otherwise.*

**Definition** **5**(CPA secure [42]). *A public-key encryption scheme Π=(Gen,Enc,Dec) has indistinguishable encryptions under a chosen plaintext attack (or is CPA secure) if for all probabilistic polynomial-time adversaries A there exists a negligible function negl such that:*
$$Pr\left[{\mathrm{PubK}}_{A,\Pi }^{\mathrm{cpa}}\left(n\right)=1\right]\leq \frac{1}{2}+negl\left(n\right).$$

For properly chosen parameters n,q and error distribution χ (e.g., in NewHope setting n=512,1024, q= 12,289; χ is the central binomial of parameter k=8), RLWE-based PKE is CPA secure assuming the hardness of ring-LWE decision problem, and a concrete CPA-secure protocol was described in [43].

**Proposition** **2.**
*The revised RLWE-based PKE in Section 3.1 preserves the CPA security assuming that the standard RLWE-based PKE with properly chosen parameters n,q and χ is CPA secure.*


**Proof.** A standard RLWE-based PKE scheme Π is CPA secure assuming the hardness of the ring-LWE decision problem, i.e., $\mathrm{Pr}\left[{\mathrm{PubK}}_{A,\Pi }^{\mathrm{cpa}}\left(n\right)=1\right]\leq \frac{1}{2}+negl\left(n\right)$. There are two modifications we made to the standard RLWE-based PKE. Firstly, at the encryption stage, Bob uses polar codes instead of uncoded plaintext. This operation has no influence on the distribution of the ciphertext and therefore preserves the security. Secondly, at the decryption step, Alice first calculates $H=\sqrt{1+\sum_{1}^{n}e_{i}^{2}+\sum_{1}^{n}s_{i}^{2}}$; then, she decides to repeat the key generation step if and only if $H>H_{\epsilon }$. Since the adversary is passive and has no idea if $H>H_{\epsilon }$ or not, he/she cannot determine if the ciphertext given to him/her is a valid one or not. Therefore, a polynomial-time adversary in the experiment ${\mathrm{PubK}}_{A,\Pi^{\prime }}^{\mathrm{cpa}}\left(n\right)$ behaves no better than in the experiment ${\mathrm{PubK}}_{A,\Pi }^{\mathrm{cpa}}\left(n\right)$, i.e.,
$${\mathrm{PubK}}_{A,\Pi^{\prime }}^{\mathrm{cpa}}\left(n\right)\leq {\mathrm{PubK}}_{A,\Pi }^{\mathrm{cpa}}\left(n\right)\leq \frac{1}{2}+negl\left(n\right).$$
□

### 3.3. Polar Encoding and SC Decoding for RLWE Channel Using Outage

In this section, we show how Bob constructs polar codes using outage at the encryption step and how Alice performs decoding at the decryption step. Denote by W:X→Y the $N(0,H^{2}r^{2})$ mod 2Z channel and by $W^{\prime }:X\to Y$ its degradation $N(0,H_{\epsilon }^{2}r^{2})$ mod 2Z channel. Given the channel degradation relationship, one is able to construct polar codes for $W^{\prime }$ and apply it to *W* in reality. Recall in Section 2.3 that the first step to construct polar codes is to calculate the Bhattacharyya parameters for every synthesized channel $W_{N}^{\prime (i)}$ for i=1,⋯,N. However, as mentioned in Section 2.3, a practical solution to calculate $Z(W_{N}^{\prime (i)})$ is to firstly quantize the continuous output alphabet of $W^{\prime }$, then construct an approximate channel of the synthesized channel at each recursion of Arıkan’s transform [33,34]. This solution proceeds as follows.

We define the likelihood ratio of $W^{\prime }$ as:(15)$$\begin{array}{c} \lambda \left(y\right):=\frac{W_{Y|X}^{\prime }\left(y|0\right)}{W_{Y|X}^{\prime }\left(y|\left\lfloor \frac{q}{2}\right\rfloor \right)},\;y\in \left[0,q\right). \end{array}$$

Since $N(0,h_{\epsilon }^{2}r^{2})$ mod 2Z is stochastically equivalent to 2Z-periodic additive Gaussian noise with variance $h_{\epsilon }^{2}r^{2}$, the transition probability $W_{Y|X}^{\prime }$ is defined as:$$\begin{array}{cc} W_{Y|X}^{\prime }\left(y|0\right) & =\sum_{\lambda \in qZ}g_{0,h_{\epsilon }^{2}r^{2}}\left(y+\lambda \right)\\ W_{Y|X}^{\prime }\left(y|\left\lfloor \frac{q}{2}\right\rfloor \right) & =\sum_{\lambda \in qZ}g_{\left\lfloor \frac{q}{2}\right\rfloor ,h_{\epsilon }^{2}r^{2}}\left(y+\lambda \right), \end{array}$$
where $g_{a,b^{2}}\left(x\right)$ is the density function of the Gaussian noise with mean *a* and variance $b^{2}$.

The channel $W^{\prime }$ is symmetric because there exists a permutation $\pi \left(y\right)=(\left\lfloor \frac{q}{2}\right\rfloor -y)\;\mathrm{mod}\;q$ such that $W^{\prime }\left(y|0\right)=W^{\prime }\left(\pi \left(y\right)|\left\lfloor \frac{q}{2}\right\rfloor \right)$. Intuitively, a symmetric channel with binary input and continuous output can be seen as a combination of infinite binary symmetric channels (BSCs). If we focus on the likelihood ratio λ(y)≥1, the crossover probability of any one of these BSCs is $\frac{1}{\lambda (y)}$. The capacity of this BSC is:$$C\left[\lambda \left(y\right)\right]=1-\frac{\lambda (y)}{\lambda (y)+1}\log_{2}\frac{\lambda (y)+1}{\lambda (y)}-\frac{1}{\lambda (y)+1}\log_{2}\left(\lambda \left(y\right)+1\right),\;\lambda \left(y\right)\geq 1.$$

If we ignore the minor geometrical error introduced by rounding operation ⌊·⌋, we observe that the intervals satisfying λ(y)≥1 is:$$A:=\left[0,\left\lfloor \frac{q}{2}\right\rfloor \right]\cup \left[q-\left\lfloor \frac{q}{2}\right\rfloor ,q\right].$$

Because C[λ(y)] is a strict monotonic function of λ(y), we divide *A* into ν segments such that for j∈{1,⋯ν}:(16)$$\begin{array}{cc} A_{j} & =\left\{y\in A:\frac{j-1}{\nu }\leq C\left[\lambda \left(y\right)\right]\leq \frac{j}{\nu }\right\}\\ \; & =\left\{y\in A:\frac{1}{h_{2}^{-1}\left(\frac{\nu -i+1}{\nu }\right)}-1\leq \lambda \left(y\right)<\frac{1}{h_{2}^{-1}\left(\frac{\nu -i}{\nu }\right)}-1\right\}, \end{array}$$
where $h_{2}\left(\cdot \right)$ is the entropy function of a Bernoulli random variable. Each $A_{j}$ corresponds to a BSC channel with crossover probability:(17)$$\begin{array}{c} p_{j}=\frac{\int_{A_{j}}W_{Y|X}^{\prime }\left(y|\left\lfloor \frac{q}{2}\right\rfloor \right)dy}{\int_{A_{j}}W_{Y|X}^{\prime }\left(y|\left\lfloor \frac{q}{2}\right\rfloor \right)dy+\int_{A_{j}}W_{Y|X}^{\prime }\left(y|0\right)dy}\;, \end{array}$$
where:$$\begin{array}{cc} \int_{A_{j}}W_{Y|X}^{\prime }\left(y|0\right)dy & =\int_{A_{j}}\sum_{\lambda \in qZ}g_{0,{\left(h_{\epsilon }r\right)}^{2}}\left(y+\lambda \right)dy\\ \int_{A_{j}}W_{Y|X}^{\prime }\left(y|\left\lfloor \frac{q}{2}\right\rfloor \right)dy & =\int_{A_{j}}\sum_{\lambda \in \lambda \in qZ}g_{\left\lfloor \frac{q}{2}\right\rfloor ,{\left(h_{\epsilon }r\right)}^{2}}\left(y+\lambda \right)dy. \end{array}$$
If we define $z_{j}$ and its conjugate ${\bar{z}}_{j}$ to be the channel output of the BSC associated with $A_{j}$, we obtain the quantized output alphabet of $W^{\prime }$ as:$$Z:=\{z_{1},{\bar{z}}_{1},z_{2},{\bar{z}}_{2},\cdots ,z_{\nu },{\bar{z}}_{\nu }\}.$$
If we denote by $W_{Q}^{\prime }$ the quantized version of the channel $W^{\prime }$, the output alphabet of $W_{Q}^{\prime }$ is $Z:=\{z_{1},{\bar{z}}_{1},\cdots ,z_{\nu },{\bar{z}}_{\nu }\}$. The following lemma claims that $W_{Q}^{\prime }$ is degraded with respect to $W^{\prime }$.

**Lemma** **4.**
*The channel $W_{Q}^{\prime }:X\to Z$ is degraded with respect to $W^{\prime }$.*


**Proof.** We supply an intermediate channel $W_{P}:Y\to Z$ such that:
$$\begin{array}{c} W_{P}\left(z|y\right)=\left\{\begin{array}{cc} \begin{array}{c} 1,\\ 1,\\ 0, \end{array} & \begin{array}{c} \mathrm{if}\;z=z_{j},\;y\in A_{j},\\ \mathrm{if}\;z={\bar{z}}_{j},\pi \left(y\right)\in A_{j},\\ \mathrm{otherwise}. \end{array} \end{array}\right. \end{array}$$
We can find that there exits a channel degradation relationship in the sense that:
$$W_{Q}^{\prime }\left(z|x\right)=\int W_{Y|X}^{\prime }\left(y|x\right)W_{P}\left(z|y\right)dy.$$
□

Now, we have a degraded version of $W^{\prime }$ with a finite output alphabet. Next, we apply Arıkan’s transform recursively to $W_{Q}^{\prime }$ and calculate the $Z(W_{QN}^{\prime (i)})$. As the channel-combining and -splitting processes continue, the alphabet size of the synthesized channels $W_{QN}^{\prime (i)}$ will increase exponentially as the recursion proceeds. To handle this problem, we employed a merging technique proposed in [34], which can reduce the alphabet size of a BDMS channel with negligible and traceable loss of performance. Specifically, a BDMS channel $W_{Q}^{\prime }$ gives rise to BDMS synthesized channels under Arıkan’s transform [32]. Any BDMS channel can be seen as a combination of BSCs. The merging technique gives an approximation of a BDMS channel by combing some of the BSCs of which it is comprised. In other words, merging approximates a BDMS channel with less BSCs, therefore a smaller output alphabet. Applying merging to the synthesized channels derived after every recursion of Arıkan’s transform can effectively restrict the output alphabet. In this manner, we can approximate the synthesized channels $W_{QN}^{\prime (i)}$ with an output alphabet no larger than a predetermined value. This makes calculating $Z(W_{QN}^{\prime (i)})$ feasible.

After we finish computing the Bhattacharyya parameters of all the $W_{QN}^{\prime (i)}$, we can define the information set A and frozen set $A^{c}$. We construct the polar codewords as:(18)$$x^{1:N}=u_{A}G_{N}\left(A\right)⊕u_{A^{c}}G_{N}\left(A^{c}\right).$$
Upon observing the channel output $y^{1:N}$, the recipient, Alice, invokes her knowledge of the CSI *h* and decides to apply the decoding or to restart the protocol. The successive cancellation (SC) decoder calculates the likelihood ratio of every synthesized channel and gives an estimation of $u_{A}$ according to the decision function:(19)$$\begin{array}{c} {\bar{u}}^{\left(i\right)}=\left\{\begin{array}{cc} \begin{array}{c} 0,\\ 1, \end{array} & \begin{array}{c} \mathrm{if}\;L_{N}^{\left(i\right)}\left(y^{1:N},{\bar{u}}^{1:i-1}\right)\geq 1\\ \mathrm{otherwise} \end{array} \end{array}\right., \end{array}$$
where the likelihood ratio $L_{N}^{\left(i\right)}\left(y^{1:N},{\bar{u}}^{1:i-1}\right)≜\frac{W_{N}^{\left(i\right)}\left(y^{1:N},{\bar{u}}^{1:i-1}|0\right)}{W_{N}^{\left(i\right)}\left(y^{1:N},{\bar{u}}^{1:i-1}|1\right)}$ can be calculated recursively by the SC decoding algorithm in [32]. The input of SC decoder λ(y) is given as:$$\begin{array}{c} \lambda \left(y\right):=\frac{W_{Y|X}\left(y|0\right)}{W_{Y|X}\left(y|\left\lfloor \frac{q}{2}\right\rfloor \right)},\;y\in \left[0,q\right), \end{array}$$
where the transition probability $W_{Y|X}$ is defined as:$$\begin{array}{cc} W_{Y|X}\left(y|0\right) & =\sum_{\lambda \in qZ}g_{0,{\left(hr\right)}^{2}}\left(y+\lambda \right),\\ W_{Y|X}\left(y|\left\lfloor \frac{q}{2}\right\rfloor \right) & =\sum_{\lambda \in qZ}g_{\left\lfloor \frac{q}{2}\right\rfloor ,{\left(hr\right)}^{2}}\left(y+\lambda \right). \end{array}$$
A block-decoding error occurs if ${\bar{u}}^{1:N}\neq u^{1:N}$; we may interchangeably use the block error probability and DFR in this work. The complexity of both polar encoding and SC decoding is $O(N\log_{2}N)$. Additionally, both algorithms require constant steps of operations for fixed choices of K,N,A, making constant-time implementations plausible. According to Theorem 2, the block error probability $P_{e}\left(N,K,A\right)$ of SC decoding is upper bounded by the sum of $Z(W_{N}^{\left(i\right)})$. Since we have $W_{Q}^{\prime }⪯W^{\prime }⪯W$ and $W_{QN}^{\prime (i)}⪯W_{N}^{\prime (i)}⪯W_{N}^{\left(i\right)}$ according to Lemmas 1 and 2, we have:(20)$$\begin{array}{c} P_{e}\left(N,K,A\right)\leq \sum_{i\in A}Z\left(W_{NQ}^{\prime (i)}\right). \end{array}$$

## 4. Results: Decoding Performance Analysis

Theorem 2 gives the upper bound on the decoding error probability (DFR of PKE equivalently) of polar codes constructed for the $N(0,r^{2}H_{\epsilon }^{2})$ mod 2Z channel and applied to the $N(0,H^{2}r^{2})$ mod 2Z channel in reality. Figure 3 depicts the upper bound on the DFR if polar codes constructed as above are used in our revised RLWE-based PKE. In the standardization process of PQC initialized by NIST, the target DFR at code rate 1/4 is $2^{-128}$. We targeted a more conservative benchmark DFR = $2^{-140}$ as was used in [15,18]. Similar to NewHope, which employs a central binomial distribution with parameter *k* to approximate the discrete Gaussian distribution (The variance of central binomial distribution is k/2, and the variance of a discrete Gaussian distribution is $r^{2}$. When calculating the upper bound on the DFR, we used a continuous Gaussian distribution instead of its discrete version to ease the analysis. However, we used the central binomial of the same variance in the experiments in Figure 4), we used the parameter $k=2r^{2}$ to denote different distributions χ from which $e,t,s,e^{\prime },e^{\prime \prime }$ were drawn. We observed that by using our polar coding scheme, we could achieve the target DFR of $2^{-140}$ for *k* as large as 55, which is significantly larger than the current choice k=8 in NewHope. A larger *k* benefits the security level of the overall scheme. Please note that schemes as NewHope compress the ciphertext before sending it out, which leads to additional compression noise. However, in this work, we only focused on the additive noise in the channel model.

The advantages of the RLWE channel model with outage are concluded as follows. Firstly, we employed an “independence” assumption so that we derived a group of i.i.d. channels. This is actually a relaxed assumption compared to the one in [15]. For example, the polynomial product e·t has correlated coefficients because of the polynomial convolution. However, we resolved the correlation produced by *e* by seeing it as a constant fading coefficient *H* over exactly one code block. The correlation left in our channel model only comes from *t*.

Secondly, the decoder is able to exploit the CSI, while the encoder makes use of the knowledge of CDI. This benefits the decoding performance significantly if compared to coding schemes that take the residue additive error term as a whole. Thirdly, the channel degradation relation makes the polar codes constructed for the degraded channel precisely fit in with the real channel. We verify our polar coding scheme in RLWE-based PKE by simulation in Figure 4. The dotted lines are the experimental results of the DFR, and the solid lines are the DFR upper bounds. At least for reasonably large code rates, the simulation results verified our estimation of the upper bounds, whereas the performance at the target code rate 1/4 was unable to be experimentally checked.

## 5. Discussion

### 5.1. Security Improvement

The new DFR margin can be exploited to increase the Gaussian noise parameter *r* (or the central binomial parameter $k=2r^{2}$) such that the security level is increased and the DFR requirement is properly satisfied. In Table 1, we illustrate to what extent the security of RLWE-based PKE was improved for n=1024,q= 12,289 compared to NewHope Round 2 if different error-correcting codes and schemes are employed. As in [15,18], a conservative target DFR was selected to be $2^{-140}$. The concrete security analysis of RLWE-based PKE, so far, has been based on the hardness of LWE [22]. The security level was estimated at the cost of primal attack and dual attack (The security estimator is available at https://github.com/tpoeppelmann/newhope (accessed on 3 March 2021)).

It was observed that the polar coding scheme described in this work gives significant security improvement compared to the one in the concurrent work using polar codes [19]. We acquired this security gain because we used the original constellation diagram $\left\{0,\;\left\lfloor \frac{q}{2}\right\rfloor \right\}$ rather than the closer and tailored one in [19]. Furthermore, our polar coding scheme gives a security improvement as attractive as the state-of-the-art record of 31.76% in [15], which employed nonconstant-time BCH and LDPC codes.

### 5.2. Constant-Time Implementation

When applying modern error-correcting codes to RLWE-based PKE, we should always be careful if the encoding and decoding enables constant-time implementations. BCH code has a good error correction capability, but its decoding proceeds in two steps: (a) locate the errors by calculating the syndrome, and (b) correct the errors if there are t/2 or fewer errors where *t* is the code distance. This is obviously not a constant-time design. LDPC code also has nonconstant-time decoding because the decoding procedure is iterative and it comes to an end when either a correct codeword is found or the maximum number of iterations is reached. Unlike the error-correcting codes (e.g., BCH, LDPC) adopted by RLWE-based PKE in the literature [15], the encoding and decoding of polar codes intrinsically enable constant-time implementations.

As for the encoding, one calculates the Bhattacharyya parameters $Z(W_{N}^{\left(i\right)})$ first and then carries out the encoding function as in Formula (Equation 18) (see Section 3.3). The most time-consuming step is to calculate $Z(W_{N}^{\left(i\right)})$; however, this can be performed offline once for all as far as the channel model in Formula (Equation 14) is known (i.e., the RLWE PKE parameters n,q, the error distribution χ, and the code rate are known). The encoding step is carried out online, and it consists of exactly $N/2\log_{2}N$ many XOR gates. An example of polar encoding for code length N=8 is given in Figure 5. It can be concluded that the mod-2 additions of polar encoding are only related to the code length *N*, and therefore, a constant-time implementation is feasible for any fixed *N*.

As for polar decoding, the running time does not vary with different actualization of the message *m* or error term drawn from χ, as is not the case for BCH and LDPC. Given the RLWE channel output $y^{1:N}$ derived from decryption, the SC decoder recursively calculates $L_{N}^{\left(i\right)}\left(y^{1:N},{\bar{u}}^{1:i-1}\right)$ and recovers the message $u^{1:N}$ according to Formula (Equation 19). The LR calculations dominate the overall complexity of decoding, which is described in Appendix A, as well as an example for code length N=8. It can be concluded that for any fixed code length *N* (N= parameter *n* of RLWE-based PKE), the SC decoding require exactly $N*\log_{2}N$ steps of LR calculations as in Formulas (Equation 21) and (Equation 22) no matter what other parameters q,χ and the code rate are. In addition, the decision-making step in Formula (Equation 19) is also constant-time because the information set A is uniquely determined by the channel model in Formula (Equation 14) and the parameters n,q,χ and code rate.

### 5.3. Complexity and Communication Overhead

Compared with the repetition codes in NewHope Round 2 [43], the proposed polar encoding and decoding scheme will for sure significantly increase the complexity. We, in this paper, mainly focused on the DFR performance and security improvement while benchmarks of the proposed scheme are not provided. Nonetheless, seeing that LDPC codes have much higher complexity than polar codes at a relatively low code rate as is explained in Appendix B (also see [44]), polar encoding and SC decoding will incur a much smaller complexity increase compared to that of 650% for LDPC, as given in [15].

Since Alice, the recipient, calculates $H=\sqrt{1+\sum_{1}^{n}e_{i}^{2}+\sum_{1}^{n}s_{i}^{2}}$ and goes back to the public key generation step if $H>H_{\epsilon }$, the averaged communication overhead is supposed to increase by a percentage of approximately ϵ for a relatively small ϵ. In this work, we set the outage probability ϵ to be a small value of 0.01, incurring a communication overhead increase by approximately 1% on average. Therefore it almost preserves the communication overhead. In addition, the proposed polar coding scheme was designed to address the additive residue noise after decryption rather than the compression noise, and we did not improve the bandwidth efficiency compared to an improvement of 5.9% and 12.8% in [18] and [15], respectively.

## 6. Conclusions

In this work, we demonstrated how to construct polar codes for RLWE-based PKE. Theoretical and numerical results were given to verify the proposed coding scheme. The motivation for doing so was to give constructive guidance on how to at least relax the “dependency” and on how to design practical and efficient error-correcting codes to lower the DFR and increase the security of RLWE-based PKE.

The pros and cons of the polar coding scheme using outage are given as follows:The polar coding scheme using outage considerably improves the error tolerance. It significantly improves the security level (measured by bits of security) of RLWE-based PKE in the NewHope setting by 28.8%, which is as attractive as the highest record in [15];The proposed polar coding scheme has lower encoding and decoding complexity at a low code rate compared to other error-correcting schemes in the literature [15]. Furthermore, it intrinsically supports constant-time implementations;Compared with the polar coding scheme in [19], this scheme is carried out in polynomial representation and uses the original modulation constellation diagram rather than the shrunk one. This avoids the trouble of switching between the polynomial and canonical representation, and the modulation space is not compromised;Since the standard process of RLWE-based PKE is amended, how it will behave under a variety of attacks is left for future work, and we proved it to be at least CPA secure nonetheless.

In conclusion, using the proposed polar coding scheme in this work, one can derive a new DFR margin and therefore improve the security of a typical RLWE-based PKE scheme (e.g., NewHope). The polar coding scheme will not increase the communication overhead. For a relatively low code rate (e.g., 0.25), polar encoding and decoding are efficient compared to other modern error-correcting codes such as LDPC. Moreover, polar codes support constant-time implementations, whereas other error-correcting codes such as LDPC and BCH do not. Future work will include a solid implementation of the proposed scheme, as well as a specific benchmarking. Besides, the hidden vulnerabilities of the proposed scheme under a variety of attacks will be investigated. 

## Figures and Tables

**Figure 1 entropy-23-00938-f001:**
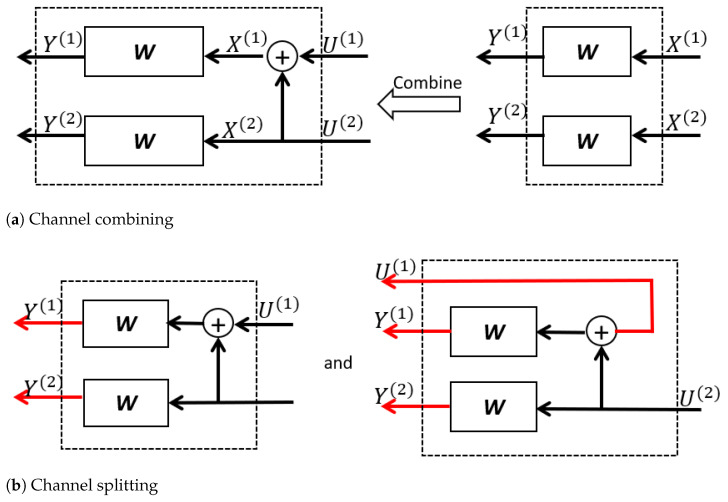
An example of channel combining and splitting for N=2.

**Figure 2 entropy-23-00938-f002:**
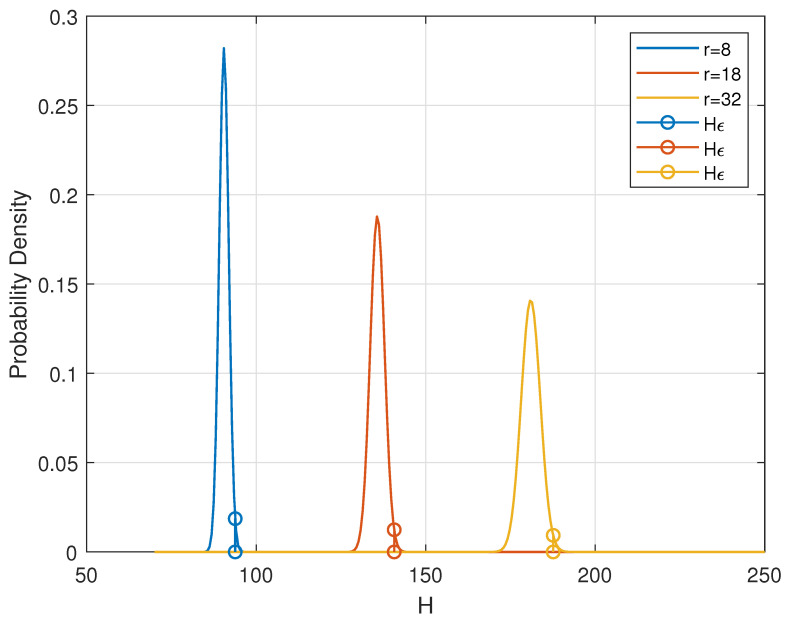
Outage probability and threshold $H_{\epsilon },\epsilon =0.01,n=1024$.

**Figure 3 entropy-23-00938-f003:**
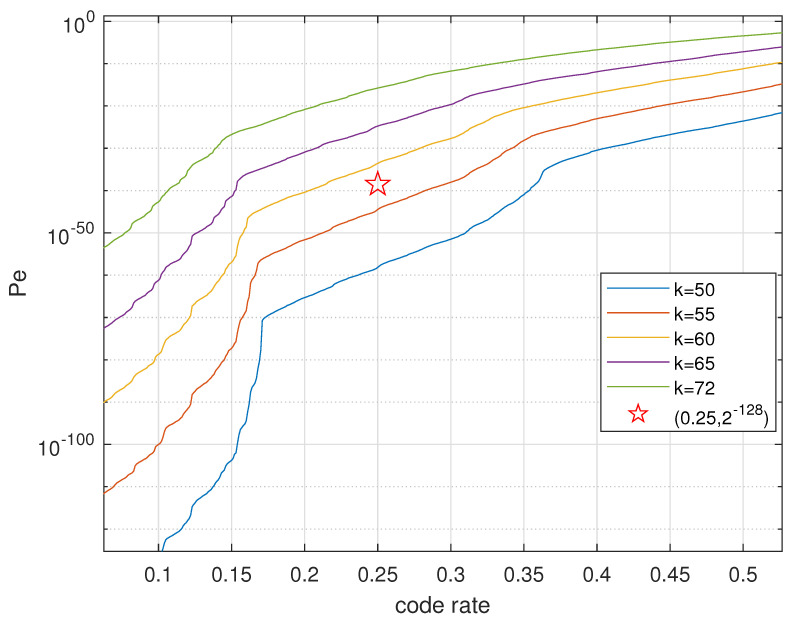
Upper bound on the frame error probability of SC decoding, ϵ=0.01,N=1024.

**Figure 4 entropy-23-00938-f004:**
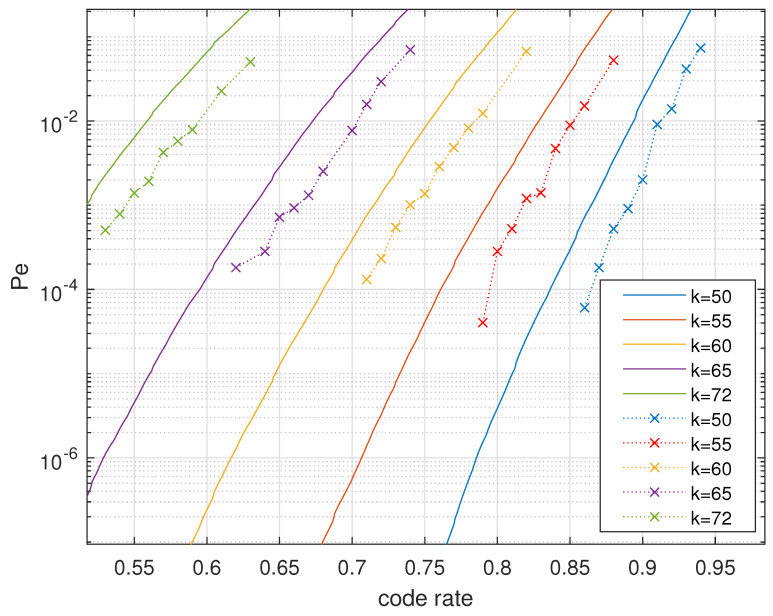
Decoding performance: analytical upper bound vs. simulation results.

**Figure 5 entropy-23-00938-f005:**
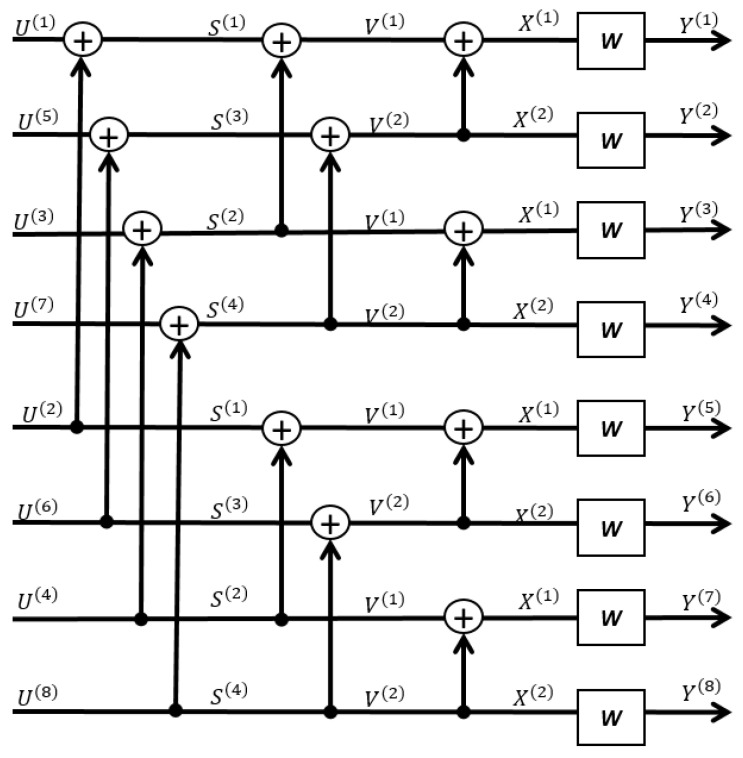
An example of polar encoding for code length N=8.

**Table 1 entropy-23-00938-t001:** Improved security level of RLWE-based PKE for n=1024,q = 12,289 using different error-correcting codes.

ECC Schemes	*k*	DFR	Classical/Quantum (bits)		Improvement
			Primal	Dual	
NewHope Round 2	8	$2^{-216}$	259/235	257/233	–
Polar codes in this work	55	$2^{-149}$	332/301	330/300	28.8%
Polar codes [19]	16	$2^{-156}$	282/256	281/255	9.4%
Song et al. [18]	14	$2^{-156}$	278/252	276/250	7.2%
Fritzmann et al. [15]	66	$2^{-140}$	341/309	338/307	31.76%

## Data Availability

Data is contained within the article.

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
