# Peer review of "How to Construct Polar Codes for Ring-LWE-Based Public Key Encryption"

_entropy, 2021, doi:10.3390/e23080938_

Round 1

Reviewer 1 Report

In this paper, public-key encryption (PKE) based on the ring learning with errors (RLWE) is considered. A very interesting approach, based on telecommunication engineering strategy, is used to construct polar codes for RLWE-based PKE.  A review of the existing results and the overview of the contributions are properly presented in Section 1, and a nice overview of preliminaries is given in Section 2 (with a focus on two important theorems).

The contribution of the paper is mostly contained in Section 3, where soundness of security is analyzed and a novel design of polar codes, based on the outage, is proposed. The presented analysis is correct, the paper is easy to follow and it definitely worth publication.

The conclusion should be extended to answer the following questions:

  • What benefits do readers can obtain from the presented analyzis?
  • What is the future work based on the foundation of this analysis?

Technical preparation is generally on a high level. However, some minor incorrectness should be removed before publication:

  • If the sentence is ended with an equation, a dot should be placed after the equation (please, see Definition 1 and Definition 2 on page 6, as well as the equation after line 244 on page 7);
  • If the equation is followed with word "where", a comma should be placed after the equation (please, see Eq. (12));
  • the text should not be placed inside the page margin (please see lines 253, 548
  • in reference [23], the first letters in the name of the journal should be capitalized;
  • in references [11] and [28], there is no need to give doi for these references only, it should be unified with the format of the other references. 

Author Response

Point 1: The conclusion should be extended to answer the following questions: What benefits do readers can obtain from the presented analyzis? What is the future work based on the foundation of this analysis?

Response 1: The conclusion section is extended as suggested to answer the two questions. For the first question, what the readers obtain is a polar coding scheme for RLWE-based PKE which can lower the DFR and improve the security without increasing the communication overhead. Polar coding will increase the complexity compared with the repetition codes in NewHope round 2 but it has lower complexity compared with LDPC for small code rates as is the case for RLWE-based PKE. Moreover, polar encoding and decoding are intrinsically constant-time algorithms as distinguishes it from other schemes like BCH and LDPC. For the second question, future work will include solid implementations, benchmarking and investigations into the security vulnerabilities of the proposed scheme.

Point 2: editing issues.

Response 2:  The editing issues pointed out are addressed accordingly.

Reviewer 2 Report

This paper proposes a method how to construct polar codes for reducing the decryption failure rate (DFR) of ring-LWE based public key encryption scheme while increasing/maintain its security. To this end, they present how to construct such polar codes and modify a ring-LWE based public key encryption scheme by employing the presented polar codes. Then, they analyze the security and DFR of the modified construction.

In general, the proposed method seems correct and provides the meaningful improvement on the security and DFR.

However, there are some doubtful parts in the paper.

  1. While the proposed scheme gives the improvements on the security and DFR, it may cause the efficiency degradation. For example, the modified encryption algorithm requires to construct polar codes and the decryption algorithm goes back to the key generation step and the whole process is restarted under a certain condition. This makes overhead and its amount needs to be compared with those of other schemes to clarify advantages/disadvantages of the proposed scheme.

  1. In the abstract, the third reason why applying previous error correcting codes to ring-LWE based public key encryption scheme is hard is to require non-constant time for syndrome decoding. However, it is unclear that the proposed construction resolves this issue. If I understand correctly, the authors only provide asymptotic complexities of the encoding and decoding algorithms, but it does not guarantee that the decoding algorithm takes constant time. This should be clearly explained.

  1. There are several typos throughout the paper. The authors need to carefully proofread and polish the manuscript.

Author Response

Response 1: We discuss the complexity and communication overhead of the proposed polar coding scheme in Section 5.3 and Appendix B. The complexity of polar encoding and decoding is higher than the repetition codes and additive threshold decoding in NewHope while it is lower than that of LDPC for small code rate like 1/4. In reference [15], the bandwidth efficiency is improved by 12.8% using LDPC and BCH codes; in reference [18], this improvement is 5.9% derived by a more accurate DFR analysis. We do not include a specific benchmarking of the proposed scheme because this work focus on how to deal with the error dependency and how to improve DFR and security. Nonetheless, considering the case where the recipient Alice goes back to the key generation step, the proposed scheme only increases the communication overhead by a percentage of approximately \epsilon where \epsilon is the outage probability. In this work, we select \epsilon=0.01. So it only incurs minor bandwidth increase.

Response 2: We further discuss the constant-time feature of polar codes in Section 5.2 and Appendix A. As to polar encoding, it takes exactly N/2*log_2(N) many XOR operations to produce a codeword of length N. As to SC decoding, it takes exactly N*log_2(N) many LR calculations according to formula (A1-A2) for any codeword of length N. As N is a public parameter, it is reasonable to say polar encoding and decoding are constant-time.

Response 3: We proofread the manuscript a few more times. Thanks for letting us know.

Round 2

Reviewer 2 Report

The authors have addressed all issues that I raised in the previous review report. Now, I support the acceptance of this manuscript.

- In Tables 2 and 3 of Appendix, units for complexity are missing. Please indicate units of them appropriately.

Author Response

It is pointed out that the complexity unit is missing in Table 2 and 3 of Appendix. For both LDPC and polar codes, the basic operation at the core of decoding is likelihood ratio (LR) calculation or equivalently LR calculation in log domain (LLR). So their complexity units (i.e., LR/LLR) are real numbers, and normally we use their floating-point representations in software implementation and fixed-point on hardware.

The complexity unit is specified in Table 2 and 3 as well as an explanation in line 688-691 on page 22.